# A Mutual Information Maximization Perspective of Language Representation Learning

**Lingpeng Kong♠, Cyprien de Masson d'Autume♠, Wang Ling♠, Lei Yu♠, Zihang Dai♡♣**
**Dani Yogatama♠**
DeepMind♠, Carnegie Mellon University♡, Google Brain♣
London, United Kingdom
{lingpenk,cyprien,lingwang,leiyu,zihangd,dyogatama}@google.com

## Abstract

We show state-of-the-art word representation learning methods maximize an objective function that is a lower bound on the mutual information between different parts of a word sequence (i.e., a sentence). Our formulation provides an alternative perspective that unifies classical word embedding models (e.g., Skip-gram) and modern contextual embeddings (e.g., BERT, XLNet). In addition to enhancing our theoretical understanding of these methods, our derivation leads to a principled framework that can be used to construct new self-supervised tasks. We provide an example by drawing inspirations from related methods based on mutual information maximization that have been successful in computer vision, and introduce a simple self-supervised objective that maximizes the mutual information between a global sentence representation and $n$-grams in the sentence. Our analysis offers a holistic view of representation learning methods to transfer knowledge and translate progress across multiple domains (e.g., natural language processing, computer vision, audio processing).

## 1 Introduction

Advances in representation learning have driven progress in natural language processing. Performance on many downstream tasks have improved considerably, achieving parity with human baselines in benchmark leaderboards such as SQuAD (Rajpurkar et al., 2016; 2018) and GLUE (Wang et al., 2019). The main ingredient is the "pretrain and fine-tune" approach, where a large text encoder is trained on an unlabeled corpus with self-supervised training objectives and used to initialize a task-specific model. Such an approach has also been shown to reduce the number of training examples that is needed to achieve good performance on the task of interest (Yogatama et al., 2019).

In contrast to first-generation models that learn word type embeddings (Mikolov et al., 2013; Pennington et al., 2014), recent methods have focused on contextual token representations—i.e., learning an encoder to represent words in context. Many of these encoders are trained with a language modeling objective, where the representation of a context is trained to be predictive of a target token by maximizing the log likelihood of predicting this token (Dai & Le, 2015; Howard & Ruder, 2018; Radford et al., 2018; 2019). In a vanilla language modeling objective, the target token is always the next token that follows the context. Peters et al. (2018) propose an improvement by adding a reverse objective that also predicts the word token that precedes the context. Following this trend, current state-of-the-art encoders such as BERT (Devlin et al., 2018) and XLNet (Yang et al., 2019) are also trained with variants of the language modeling objective: masked language modeling and permutation language modeling.

In this paper, we provide an alternative view and show that these methods also maximize a lower bound on the mutual information between different parts of a word sequence. Such a framework is inspired by the InfoMax principle (Linsker, 1988) and has been the main driver of progress in self-supervised representation learning in other domains such as computer vision, audio processing, and reinforcement learning (Belghazi et al., 2018; van den Oord et al., 2019; Hjelm et al., 2019;

Bachman et al., 2019; O'Connor & Veeling, 2019). Many of these methods are trained to maximize a particular lower bound called InfoNCE (van den Oord et al., 2019)—also known as contrastive learning (Arora et al., 2019). The main idea behind contrastive learning is to divide an input data into multiple (possibly overlapping) views and maximize the mutual information between encoded representations of these views, using views derived from other inputs as negative samples. In §2, we provide an overview of representation learning with mutual information maximization. We then show how the skip-gram objective (§3.1; Mikolov et al. 2013), masked language modeling (§3.2; Devlin et al. 2018), and permutation language modeling (§3.3; Yang et al. 2019), fit in this framework.

In addition to providing a principled theoretical understanding that bridges progress in multiple areas, our proposed framework also gives rise to a general class of word representation learning models which serves as a basis for designing and combining self-supervised training objectives to create better language representations. As an example, we show how to use this framework to construct a simple self-supervised objective that maximizes the mutual information between a sentence and $n$-grams in the sentence (§4). We combine it with a variant of the masked language modeling objective and show that the resulting representation performs better, particularly on tasks such as question answering and linguistics acceptability (§5).

## 2 MUTUAL INFORMATION MAXIMIZATION

Mutual information measures dependencies between random variables. Given two random variables $A$ and $B$, it can be understood as how much knowing $A$ reduces the uncertainty in $B$ or vice versa. Formally, the mutual information between $A$ and $B$ is:

$$I(A, B) = H(A) - H(A \mid B) = H(B) - H(B \mid A).$$

Consider $A$ and $B$ to be different views of an input data (e.g., a word and its context, two different partitions of a sentence). Consider a function $f$ that takes $A = a$ and $B = b$ as its input. The goal of training is to learn parameters of the function $f$ that maximizes $I(A, B)$.

Maximizing mutual information directly is generally intractable when the function $f$ consists of modern encoders such as neural networks (Paninski, 2003), so we need to resort to a lower bound on $I(A, B)$. One particular lower bound that has been shown to work well in practice is InfoNCE (Logeswaran & Lee, 2018; van den Oord et al., 2019),[1] which is based on Noise Contrastive Estimation (NCE; Gutmann & Hyvarinen, 2012).[2] InfoNCE is defined as:

$$I(A, B) \geq \mathbb{E}_{p(A,B)} \left[ f_{\boldsymbol{\theta}}(a, b) - \mathbb{E}_{q(\tilde{\mathcal{B}})} \left[ \log \sum_{\tilde{b} \in \tilde{\mathcal{B}}} \exp f_{\boldsymbol{\theta}}(a, \tilde{b}) \right] \right] + \log | \tilde{\mathcal{B}} |, \qquad (1)$$

where $a$ and $b$ are different views of an input sequence, $f_{\boldsymbol{\theta}} \in \mathbb{R}$ is a function parameterized by $\boldsymbol{\theta}$ (e.g., a dot product between encoded representations of a word and its context, a dot product between encoded representations of two partitions of a sentence), and $\tilde{\mathcal{B}}$ is a set of samples drawn from a proposal distribution $q(\tilde{\mathcal{B}})$. The set $\tilde{\mathcal{B}}$ contains the positive sample $b$ and $|\tilde{\mathcal{B}}| - 1$ negative samples.

Learning representations based on this objective is also known as contrastive learning. Arora et al. (2019) show representations learned by such a method have provable performance guarantees and reduce sample complexity on downstream tasks.

We note that InfoNCE is related to cross-entropy. When $\tilde{\mathcal{B}}$ always includes all possible values of the random variable $B$ (i.e., $\tilde{\mathcal{B}} = \mathcal{B}$) and they are uniformly distributed, maximizing InfoNCE is analogous to maximizing the standard cross-entropy loss:

$$\mathbb{E}_{p(A,B)} \left[ f_{\boldsymbol{\theta}}(a, b) - \log \sum_{\tilde{b} \in \mathcal{B}} \exp f_{\boldsymbol{\theta}}(a, \tilde{b}) \right]. \qquad (2)$$

---

[1]Alternative bounds include Donsker-Vardhan representation (Donsker & Varadhan, 1983) and Jensen-Shannon estimator (Nowozin et al., 2016), but we focus on InfoNCE here.

[2] See van den Oord et al. (2019); Poole et al. (2019) for detailed derivations of InfoNCE as a bound on mutual information.

Eq. 2 above shows that InfoNCE is related to maximizing $p_{\boldsymbol{\theta}}(b \mid a)$, and it approximates the summation over elements in $\mathcal{B}$ (i.e., the partition function) by negative sampling. As a function of the negative samples, the InfoNCE bound is tighter when $\tilde{\mathcal{B}}$ contains more samples (as can be seen in Eq. 1 above by inspecting the $\log |\tilde{\mathcal{B}}|$ term). Approximating a softmax over a large vocabulary with negative samples is a popular technique that has been widely used in natural language processing in the past. We discuss it here to make the connection under this framework clear.

## 3 MODELS

We describe how Skip-gram, BERT, and XLNet fit into the mutual information maximization framework as instances of InfoNCE. In the following, we assume that $f_{\boldsymbol{\theta}}(a, b) = g_{\boldsymbol{\psi}}(b)^{\top} g_{\boldsymbol{\omega}}(a)$, where $\boldsymbol{\theta} = \{\boldsymbol{\omega}, \boldsymbol{\psi}\}$. Denote the vocabulary set by $\mathcal{V}$ and the size of the vocabulary by $V$. For word representation learning, we seek to learn an encoder parameterized by $\boldsymbol{\omega}$ to represent each word in a sequence $\boldsymbol{x} = \{x_1, x_1, \ldots, x_T\}$ in $d$ dimensions. For each of the models we consider in this paper, $a$ and $b$ are formed by taking different parts of $\boldsymbol{x}$ (e.g., $a := x_0$ and $b := x_T$).

### 3.1 SKIP-GRAM

We first start with a simple word representation learning model Skip-gram (Mikolov et al., 2013). Skip-gram is a method for learning word representations that relies on the assumption that a good representation of a word should be predictive of its context. The objective function that is maximized in Skip-gram is: $\mathbb{E}_{p(x_i, x_j^i)} \left[ p(x_j^i \mid x_i) \right]$, where $x_i$ is a word token and $x_j^i$ is a context word of $x_i$.

Let $b$ be the context word to be predicted $x_j^i$ and $a$ be the input word $x_i$. Recall that $f_{\boldsymbol{\theta}}(a, b)$ is $g_{\boldsymbol{\psi}}(b)^{\top} g_{\boldsymbol{\omega}}(a)$. The skip-gram objective function can be written as an instance of InfoNCE (Eq. 1) where $g_{\boldsymbol{\psi}}(b)$ and $g_{\boldsymbol{\omega}}(a)$ are embedding lookup functions that map each word type to $\mathbb{R}^d$. (i.e., $g_{\boldsymbol{\psi}}(b), g_{\boldsymbol{\omega}}(a) : \mathcal{V} \to \mathbb{R}^d$).

$p(x_j^i \mid x_i)$ can either be computed using a standard softmax over the entire vocabulary or with negative sampling (when the vocabulary is very large). These two approaches correspond to different choices of $\tilde{\mathcal{B}}$. In the softmax approach, $\tilde{\mathcal{B}}$ is the full vocabulary set $\mathcal{V}$ and each word in $\mathcal{V}$ is uniformly distributed. In negative sampling, $\tilde{\mathcal{B}}$ is a set of negative samples drawn from e.g., a unigram distribution.

While Skip-gram has been widely accepted as an instance contrastive learning (Mikolov et al., 2013; Mnih & Kavukcuoglu, 2013), we include it here to illustrate its connection with modern approaches such as BERT and XLNet described subsequently. We can see that the two views of an input sentence that are considered in Skip-gram are two words that appear in the same sentence, and they are encoded using simple lookup functions.

### 3.2 BERT

Devlin et al. (2018) introduce two self-supervised tasks for learning contextual word representations: masked language modeling and next sentence prediction. Previous work suggests that the next sentence prediction objective is not necessary to train a high quality BERT encoder and the masked language modeling appears to be the key to learn good representations (Liu et al., 2019; Joshi et al., 2019; Lample & Conneau, 2019), so we focus on masked language modeling here. However, we also show how next sentence prediction fits into our framework in Appendix A.

In masked language modeling, given a sequence of word tokens of length $T$, $\boldsymbol{x} = \{x_1, \ldots, x_T\}$, BERT replaces 15% of the tokens in the sequence with (i) a mask symbol 80% of the time, (ii) a random word 10% of the time, or (iii) its original word. For each replaced token, it introduces a term in the masked language modeling training objective to predict the original word given the perturbed sequence $\hat{\boldsymbol{x}}_i = \{x_1, \ldots, \hat{x}_i, \ldots, x_T\}$ (i.e., the sequence $\boldsymbol{x}$ masked at $x_i$). This training objective can be written as: $\mathbb{E}_{p(x_i, \hat{\boldsymbol{x}}_i)}[p(x_i \mid \hat{\boldsymbol{x}}_i)]$.

Following our notation in §2, we have $f_{\boldsymbol{\theta}}(a, b) = g_{\boldsymbol{\psi}}(b)^{\top} g_{\boldsymbol{\omega}}(a)$. Let $b$ be a masked word $x_i$ and $a$ be the masked sequence $\hat{\boldsymbol{x}}_i$. Consider a Transformer encoder parameterized by $\boldsymbol{\omega}$ and denote $g_{\boldsymbol{\omega}}(\hat{\boldsymbol{x}}_i) \in \mathbb{R}^d$ as a function that returns the final hidden state corresponding to the $i$-th token (i.e., the masked token) after running $\hat{\boldsymbol{x}}_i$ through the Transformer. Let $g_{\boldsymbol{\psi}} : \mathcal{V} \to \mathbb{R}^d$ be a lookup function that maps each word type into a vector and $\tilde{\mathcal{B}} = \mathcal{B}$ be the full vocabulary set $\mathcal{V}$. Under this formulation, the masked language modeling objective maximizes Eq. 1 and different choices of masking probabilities can be understood as manipulating the joint distributions $p(a, b)$. In BERT, the two views of a sentence correspond to a masked word in the sentence and its masked context.

**Contextual vs. non-contextual.** It is generally understood that the main difference between Skip-gram and BERT is that Skip-gram learns representations of word types (i.e., the representation for a word is always the same regardless of the context it appears in) and BERT learns representations of word tokens. We note that under our formulation for either Skip-gram or BERT, the encoder that we want to learn appears in $g_{\boldsymbol{\omega}}$, and $g_{\boldsymbol{\psi}}$ is not used after training. We show that Skip-gram and BERT maximizes a similar objective, and the main difference is in the choice of the encoder that forms $g_{\boldsymbol{\omega}}$—a context dependent Transformer encoder that takes a sequence as its input for BERT and a simple word embedding lookup for Skip-gram.

### 3.3 XLNET

Yang et al. (2019) propose a permutation language modeling objective to learn contextual word representations. This objective considers all possible factorization permutations of a joint distribution of a sentence. Given a sentence $\boldsymbol{x} = \{x_1, \ldots, x_T\}$, there are $T!$ ways to factorize its joint distribution.[3] Given a sentence $\boldsymbol{x}$, denote a permutation by $\boldsymbol{z} \in \mathcal{Z}$. XLNet optimizes the objective function:

$$\mathbb{E}_{p(\boldsymbol{x})} \left[ \mathbb{E}_{p(\boldsymbol{z})} \left[ \sum_{t=1}^{T} \log p(x_t^{\boldsymbol{z}} \mid \boldsymbol{x}_{<t}^{\boldsymbol{z}}) \right] \right].$$

As a running example, consider a permutation order `3,1,5,2,4` for a sentence $x_1, x_2, x_3, x_4, x_5$. Given the order, XLNet is only trained to predict the last $S$ tokens in practice. For $S = 1$, the context sequence used for training is $x_1, x_2, x_3, \_, x_5$, with $x_4$ being the target word.

In addition to replacing the Transformer encoder with Transformer XL (Dai et al., 2019), a key architectural innovation of XLNet is the two-stream self-attention. In two-stream self attention, a shared encoder is used to compute two sets of hidden representations from one original sequence. They are called the query stream and the content stream. In the query stream, the input sequence is masked at the target position, whereas the content stream sees the word at the target position. Words at future positions for the permutation order under consideration are also masked in both streams. These masks are implemented as two attention mask matrices. During training, the final hidden representation for a target position from the query stream is used to predict the target word.

Since there is only one set of encoder parameters for both streams, we show that we can arrive at the permutation language modeling objective from the masked language modeling objective with an architectural change in the encoder. Denote a hidden representation by $\mathbf{h}_t^k$, where $t$ indexes the position and $k$ indexes the layer, and consider the training sequence $x_1, x_2, x_3, \_, x_5$ and the permutation order `3,1,5,2,4`. In BERT, we compute attention scores to obtain $\mathbf{h}_t^k$ from $\mathbf{h}_t^{k-1}$ for every $t$ (i.e., $t = 1, \ldots, T$), where $\mathbf{h}_4^0$ is the embedding for the mask symbol. In XLNet, the attention scores for future words in the permutation order are masked to 0. For example, when we compute $\mathbf{h}_1^k$, only the attention score from $\mathbf{h}_3^{k-1}$ is considered (since the permutation order is `3,1,5,2,4`). For $\mathbf{h}_5^k$, we use $\mathbf{h}_1^{k-1}$ and $\mathbf{h}_3^{k-1}$. XLNet does not require a mask symbol embedding since the attention score from a masked token is always zeroed out with an attention mask (implemented as a matrix). As a result, we can consider XLNet training as masked language modeling with stochastic attention masks in the encoder.

It is now straightforward to see that the permutation language modeling objective is an instance of Eq.1, where $b$ is a target token $x_i$ and $a$ is a masked sequence $\hat{\boldsymbol{x}}_i = \{x_1, \ldots, \hat{x}_i, \ldots, x_T\}$. Similar to

---

[3]For example, we can factorize $p(\boldsymbol{x}) = p(x_1)p(x_2 \mid x_1) \ldots, p(x_T \mid x_1, \ldots, x_{T-1}) = p(x_T)p(x_{T-1} \mid x_T) \ldots, p(x_1 \mid x_T, \ldots, x_2)$, and many others.

Table 1: Summary of methods as instances of contrastive learning. See text for details.

| Objective | $a$ | $b$ | $p(a, b)$ | $g_{\boldsymbol{\omega}}$ | $g_{\boldsymbol{\psi}}$ |
|---|---|---|---|---|---|
| Skip-gram | word | word | word and its context | lookup | lookup |
| MLM | context | masked word | masked tokens probability | Transformer | lookup |
| NSP | sentence | sentence | (non-)consecutive sentences | Transformer | lookup |
| XLNet | context | masked word | factorization permutation | TXL++ | lookup |
| DIM | context | masked $n$-grams | sentence and its $n$-grams | Transformer | not used |

BERT, we have a Transformer encoder parameterized by $\boldsymbol{\omega}$ and denote $g_{\boldsymbol{\omega}}(\hat{\boldsymbol{x}}_i) \in \mathbb{R}^d$ as a function that returns the final hidden state corresponding to the $i$-th token (i.e., the masked token) after running $\hat{\boldsymbol{x}}_i$ through the Transformer. Let $g_{\boldsymbol{\psi}} : \mathcal{V} \to \mathbb{R}^d$ be a lookup function that maps each word type into a vector and $\tilde{\mathcal{B}} = \mathcal{B}$ be the full vocabulary set $\mathcal{V}$. The main difference between BERT and XLNet is that the encoder that forms $g_{\boldsymbol{\omega}}$ used in XLNet implements attention masking based on a sampled permutation order when building its representations. In addition, XLNet and BERT also differ in the choice of $p(a, b)$ since each of them has its own masking procedure. However, we can see that both XLNet and BERT maximize the same objective.

## 4   INFOWORD

Our analysis on Skip-Gram, BERT, and XLNet shows that their objective functions are different instances of InfoNCE in Eq.1, although they are typically trained using the entire vocabulary set for $\tilde{\mathcal{B}}$ instead of negative sampling. These methods differ in how they choose which views of a sentence they use as $a$ and $b$, the data distribution $p(a, b)$, and the architecture of the encoder for computing $g_{\boldsymbol{\omega}}$, which we summarize in Table 1. Seen under this unifying framework, we can observe that progress in the field has largely been driven by using a more powerful encoder to represent $g_{\boldsymbol{\omega}}$. While we only provide derivations for Skip-gram, BERT, and XLNet, it is straightforward to show that other language-modeling-based pretraining-objectives such as those used in ELMo (Peters et al., 2018) and GPT-2 (Radford et al., 2019) can be formulated under this framework.

Our framework also allows us to draw connections to other mutual information maximization representation learning methods that have been successful in other domains (e.g., computer vision, audio processing, reinforcement learning). In this section, we discuss an example derive insights to design a simple self-supervised objective for learning better language representations.

Deep InfoMax (DIM; Hjelm et al., 2019) is a mutual information maximization based representation learning method for images. DIM shows that maximizing the mutual information between an image representation and local regions of the image improves the quality of the representation. The complete objective function that DIM maximizes consists of multiple terms. Here, we focus on a term in the objective that maximizes the mutual information between local features and global features. We describe the main idea of this objective for learning representations from a one-dimensional sequence, although it is originally proposed to learn from a two-dimensional object.

Given a sequence $\boldsymbol{x} = \{x_1, x_2, \ldots, x_T\}$, we consider the "global" representation of the sequence to be the hidden state of the first token (assumed to be a special start of sentence symbol) after contextually encoding the sequence $g_{\boldsymbol{\omega}}(\boldsymbol{x})$,[4] and the local representations to be the encoded representations of each word in the sequence $g_{\boldsymbol{\psi}}(x_t)$. We can use the contrastive learning framework to design a task that maximizes the mutual information between this global representation vector and its corresponding "local" representations using local representations from other sequences $g_{\boldsymbol{\psi}}(\hat{x}_t)$ as negative samples. This is analogous to training the global representation vector of a sentence to *choose* which words appear in the sentence and which words are from other sentences.[5] However, if we feed the original sequence $\boldsymbol{x}$ to the encoder and take the hidden state of the first token as the global

---

[4]Alternatively, the global representation can be the averaged representations of words in the sequence although we do not explore this in our experiments.

[5] We can see that this self-supervised task is related to the next sentence prediction objective in BERT. However, instead of learning a global representation (assumed to be the representation of the first token in BERT)

representation, the task becomes trivial since the global representation is built using all the words in the sequence. We instead use a masked sequence $a := \hat{\boldsymbol{x}}_t = \{x_1, \ldots, \hat{x}_t, \ldots, x_T\}$ and $b := x_t$.

State-of-the-art methods based on language modeling objectives consider all negative samples since the second view of the input data (i.e., the part denoted by $b$ in Eq. 1) that are used is simple and it consists of only a target word—hence the size of the negative set is still manageable. A major benefit of the contrastive learning framework is that we only need to be able to take negative samples for training. Instead of individual words, we can use $n$-grams as the local representations.[6] Denote an $n$-gram by $\boldsymbol{x}_{i:j}$ and a masked sequence masked at position $i$ to $j$ by $\hat{\boldsymbol{x}}_{i:j}$ We define $\mathcal{J}_{\text{DIM}}$ as:

$$\mathcal{J}_{\text{DIM}} = \mathbb{E}_{p(\hat{\boldsymbol{x}}_{i:j}, \boldsymbol{x}_{i:j})} \left[ g_{\boldsymbol{\omega}}(\hat{\boldsymbol{x}}_{i:j})^\top g_{\boldsymbol{\omega}}(\boldsymbol{x}_{i:j}) - \log \sum_{\tilde{\boldsymbol{x}}_{i:j} \in \tilde{\mathcal{S}}} \exp(g_{\boldsymbol{\omega}}(\hat{\boldsymbol{x}}_{i:j})^\top g_{\boldsymbol{\omega}}(\tilde{\boldsymbol{x}}_{i:j})) \right],$$

where $\hat{\boldsymbol{x}}_{i:j}$ is a sentence masked at position $i$ to $j$, $\boldsymbol{x}_{i:j}$ is an $n$-gram spanning from $i$ to $j$, and $\tilde{\boldsymbol{x}}_{i:j}$ is an $n$-gram from a set $\tilde{\mathcal{S}}$ that consists of the positive sample $\boldsymbol{x}_{i:j}$ and negative $n$-grams from other sentences in the corpus. We use one Transformer to encode both views, so we do not need $g_{\boldsymbol{\psi}}$ here.

Since the main goal of representation learning is to train an encoder parameterized by $\boldsymbol{\omega}$, it is possible to combine multiple self-supervised tasks into an objective function in the contrastive learning framework. Our model, which we denote INFOWORD, combines the above objective—which is designed to improve sentence and span representations—with a masked language modeling objective $\mathcal{J}_{\text{MLM}}$ for learning word representations. The only difference between our masked language modeling objective and the standard masked language modeling objective is that we use negative sampling to construct $\tilde{\mathcal{V}}$ by sampling from the unigram distribution. We have:

$$\mathcal{J}_{\text{MLM}} = \mathbb{E}_{p(\hat{\boldsymbol{x}}_i, x_i)} \left[ g_{\boldsymbol{\omega}}(\hat{\boldsymbol{x}}_i)^\top g_{\boldsymbol{\psi}}(x_i) - \log \sum_{\tilde{x}_i \in \tilde{\mathcal{V}}} \exp(g_{\boldsymbol{\omega}}(\hat{\boldsymbol{x}}_i)^\top g_{\boldsymbol{\psi}}(\tilde{x}_i)) \right],$$

where $\hat{\boldsymbol{x}}_i$ a sentence masked at position $i$ and $x_i$ is the $i$-th token in the sentence.

Our overall objective function is a weighted combination of the two terms above:

$$\mathcal{J}_{\text{INFOWORD}} = \lambda_{\text{MLM}} \mathcal{J}_{\text{MLM}} + \lambda_{\text{DIM}} \mathcal{J}_{\text{DIM}},$$

where $\lambda_{\text{MLM}}$ and $\lambda_{\text{DIM}}$ are hyperparameters that balance the contribution of each term.

## 5 EXPERIMENTS

In this section, we evaluate the effects of training masked language modeling with negative sampling and adding $\mathcal{J}_{\text{DIM}}$ to the quality of learned representations.

### 5.1 SETUP

We largely follow the same experimental setup as the original BERT model (Devlin et al., 2018). We have two Transformer architectures similar to BERT$_{\text{BASE}}$ and BERT$_{\text{LARGE}}$. BERT$_{\text{BASE}}$ has 12 hidden layers, 768 hidden dimensions, and 12 attention heads (110 million parameters); whereas BERT$_{\text{LARGE}}$ has 24 hidden layers, 1024 hidden dimensions, and 16 attention heads (340 million parameters).

For each of the Transformer variant above, we compare three models in our experiments:

- BERT: The original BERT model publicly available in `https://github.com/google-research/bert`.
- BERT-NCE: Our reimplementation of BERT. It differs from the original implementation in several ways: (1) we only use the masked language modeling objective and remove next sentence prediction, (2) we use negative sampling instead of softmax, and (3) we only use one sentence for each training example in a batch.

---

to be predictive of whether two sentences are consecutive sentences, it learns its global representation to be predictive of words in the original sentence.

[6]Local image patches used in DIM are analogous to $n$-grams in a sentence.

- INFOWORD: Our model described in §4. The main difference between INFOWORD and BERT-NCE is the addition of $\mathcal{I}_{\text{DIM}}$ to the objective function. We discuss how we mask the data for $\mathcal{I}_{\text{DIM}}$ in §5.2.

## 5.2 PRETRAINING

We use the same training corpora and apply the same preprocessing and tokenization as BERT. We create masked sequences for training with $\mathcal{I}_{\text{DIM}}$ as follows. We iteratively sample $n$-grams from a sequence until the masking budget (15% of the sequence length) has been spent. At each sampling iteration, we first sample the length of the $n$-gram (i.e., $n$ in $n$-grams) from a Gaussian distribution $\mathcal{N}(5, 1)$ clipped at 1 (minimum length) and 10 (maximum length). Since BERT tokenizes words into subwords, we measure the $n$-gram length at the word level and compute the masking budget at the subword level. This procedure is inspired by the masking approach in Joshi et al. (2019).

For negative sampling, we use words and $n$-grams from other sequences in the same batch as negative samples (for MLM and DIM respectively). There are approximately 70,000 subwords and 10,000 $n$-grams (words and phrases) in a batch. We discuss hyperparameter details in Appendix B.

## 5.3 FINE-TUNING

We evaluate on two benchmarks: GLUE (Wang et al., 2019) and SQuAD(Rajpurkar et al., 2016). We train a task-specific decoder and fine-tune pretrained models for each dataset that we consider. We describe hyperparameter details in Appendix B.

**GLUE** is a set of natural language understanding tasks that includes sentiment analysis, linguistic acceptability, paraphrasing, and natural language inference. Each task is formulated as a classification task. The tasks in GLUE are either a single-sentence classification task or a sentence pair classification task. We follow the same setup as the original BERT model and add a start of sentence symbol (i.e., the `CLS` symbol) to every example and use a separator symbol (i.e., the `SEP` symbol) to separate two concatenated sentences (for sentence pair classification tasks). We add a linear transformation and a softmax layer to predict the correct label (class) from the representation of the first token of the sequence.

**SQuAD** is a reading comprehension dataset constructed from Wikipedia articles. We report results on SQuAD 1.1. Here, we also follow the same setup as the original BERT model and predict an answer span—the start and end indices of the correct answer in the context. We use a standard span predictor as the decoder, which we describe in details in Appendix C.

## 5.4 RESULTS

We show our main results in Table 2 and Table 3. Our BERT reimplementation with negative sampling underperforms the original BERT model on GLUE but is significantly better on SQuAD. However, we think that the main reasons for this performance discrepancy are the different masking procedures (we use span-based masking instead of whole-word masking) and the different ways training examples are presented to the model (we use one consecutive sequence instead of two sequences separated by the separator symbol). Comparing BERT-NCE and INFOWORD, we observe the benefit of the new self-supervised objective $\mathcal{I}_{\text{DIM}}$ (better overall GLUE and SQuAD results), particularly on tasks such as question answering and linguistics acceptability that seem to require understanding of longer phrases. In order to better understand our model, we investigate its performance with varying numbers of training examples and different values of $\lambda_{\text{DIM}}$ on the SQuAD development set and show the results in Figure 1 (for models with the BASE configuration). We can see that INFOWORD consistently outperforms BERT-NCE and the performance gap is biggest when the dataset is smallest, suggesting the benefit of having better pretrained representations when there are fewer training examples.

Table 2: Summary of results on GLUE.

| | Model | CoLA | SST-2 | MRPC | QQP | MNLI (M/MM) | QNLI | RTE | GLUE AVG |
|---|---|---|---|---|---|---|---|---|---|
| BASE | BERT | 52.1 | 93.5 | 88.9 | 71.2 | 84.6/83.4 | 90.5 | 66.4 | 78.8 |
| | BERT-NCE | 50.8 | 93.0 | 88.6 | 70.5 | 83.2/83.0 | 90.9 | 65.9 | 78.2 |
| | INFOWORD | 53.3 | 92.5 | 88.7 | 71.0 | 83.7/82.4 | 91.4 | 68.3 | **78.9** |
| LARGE | BERT | 60.5 | 94.9 | 89.3 | 72.1 | 86.7/85.9 | 92.7 | 70.1 | **81.5** |
| | BERT-NCE | 54.7 | 93.1 | 89.5 | 71.2 | 85.8/85.0 | 92.7 | 72.5 | 80.6 |
| | INFOWORD | 57.5 | 94.2 | 90.2 | 71.3 | 85.8/84.8 | 92.6 | 72.0 | 81.1 |

Table 3: Summary of results on SQuAD 1.1.

| | Model | DEV $F_1$ | DEV EM | TEST $F_1$ | TEST EM |
|---|---|---|---|---|---|
| BASE | BERT | 88.5 | 80.8 | - | - |
| | BERT-NCE | 90.2 | 83.3 | 90.9 | 84.4 |
| | INFOWORD | **90.7** | **84.0** | **91.4** | **84.7** |
| LARGE | BERT | 90.9 | 84.1 | 91.3 | 84.3 |
| | BERT-NCE | 92.0 | 85.9 | 92.7 | 86.6 |
| | INFOWORD | **92.6** | **86.6** | **93.1** | **87.3** |

## 5.5 DISCUSSION

**Span-based models.** We show how to design a simple self-supervised task in the InfoNCE framework that improves downstream performance on several datasets. Learning language representations to predict contiguous masked tokens has been explored in other context, and the objective introduced in $\mathcal{J}_{\mathrm{DIM}}$ is related to these span-based models such as SpanBERT (Joshi et al., 2019) and MASS (Song et al., 2019). While our experimental goal is to demonstrate the benefit of contrastive learning for constructing self-supervised tasks, we note that INFOWORD is simpler to train and exhibits similar trends to SpanBERT that outperforms baseline models. We leave exhaustive comparisons to these methods to future work.

**Mutual information maximization.** A recent study has questioned whether the success of InfoNCE as an objective function is due to its property as a lower bound on mutual information and provides an alternative hypothesis based on metric learning (Tschannen et al., 2019). Regardless of the prevailing perspective, InfoNCE is widely accepted as a good representation learning objective, and formulating state-of-the-art language representation learning methods under this framework offers valuable insights that unifies many popular representation learning methods.

**Regularization.** Image representation learning methods often incorporate a regularization term in its objective function to encourage learned representations to look like a prior distribution (Hjelm et al., 2019; Bachman et al., 2019). This is useful for incorporating prior knowledge into a representation learning model. For example, the DeepInfoMax model has a term in its objective that encourages the learned representation from the encoder to match a uniform prior. Regularization is not commonly used when learning language representations. Our analysis and the connection we draw to representation learning methods used in other domains provide an insight into possible ways to incorporate prior knowledge into language representation learning models.

**Future directions.** The InfoNCE framework provides a holistic way to view progress in language representation learning. The framework is very flexible and suggests several directions that can be explored to improve existing methods. We show that progress in the field has been largely driven by innovations in the encoder which forms $g_{\boldsymbol{\omega}}$. InfoNCE is based on maximizing the mutual information between different views of an input data, and it facilitates training on structured views as long as we can perform negative sampling (van den Oord et al., 2019; Bachman et al., 2019). Our analysis demonstrates that existing methods based on language modeling objectives only consider a single target word as one of the views. We think that incorporating more complex views (e.g., higher-order or skip $n$-grams, syntactic and semantic parses, etc.) and designing appropriate self-supervised tasks

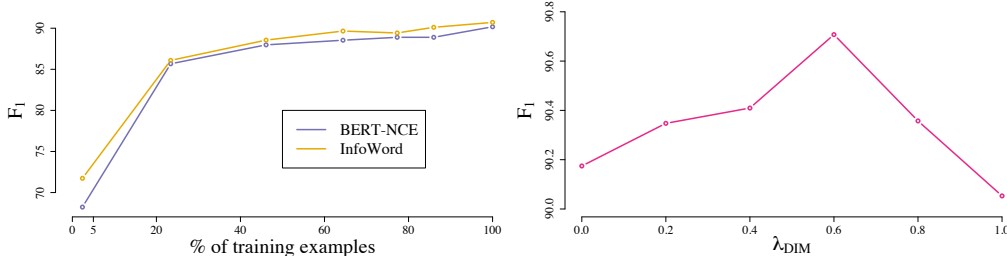

Figure 1: The left plot shows $F_1$ scores of BERT-NCE and INFOWORD as we increase the percentage of training examples on SQuAD (dev). The right plot shows $F_1$ scores of INFOWORD on SQuAD (dev) as a function of $\lambda_{\text{DIM}}$.

is a promising future direction. A related area that is also underexplored is designing methods to obtain better negative samples.

## 6  CONCLUSION

We analyzed state-of-the-art language representation learning methods from the perspective of mutual information maximization. We provided a unifying view of classical and modern word embedding models and showed how they relate to popular representation learning methods used in other domains. We used this framework to construct a new self-supervised task based on maximizing the mutual information between the global representation and local representations of a sentence. We demonstrated the benefit of this new task via experiments on GLUE and SQuAD.

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

## A    NEXT SENTENCE PREDICTION

We show that the next sentence prediction objective used in BERT is an instance of contrastive learning in this section. In next sentence prediction, given two sentences $x^1$ and $x^2$, the task is to predict whether these are two consecutive sentences or not. Training data for this task is created by sampling a random second sentence $\hat{x}^2$ from the corpus to be used as a negative example 50% of the time.

Consider a discriminator (i.e., a classifier with parameters $\phi$) that takes encoded representations of concatenated $x^1$ and $x^2$ and returns a score. We denote this discriminator by $d_\phi(x^1, x^2)$. The next sentence prediction objective function is:

$$\mathbb{E}_{p(x^1, x^2)} \left[ \log d_\phi(g_\omega([x^1, x^2])) + \log(1 - d_\phi(g_\omega([x^1, \tilde{x}^2]))) \right].$$

This objective function—which is used for training BERT—is known in the literature as "local" Noise Contrastive Estimation (Gutmann & Hyvarinen, 2012). Since summing over all possible negative sentences is intractable, BERT approximates this by using a binary classifier to distinguish real samples and noisy samples.

An alternative approximation to using a binary classifier is to use "global NCE", which is what InfoNCE is based on. Here, we have:

$$\mathbb{E}_{p(x^1, x^2)} \left[ \psi^\top g_\omega([x^1, x^2]) - \log \sum_{\tilde{x}^2 \in \tilde{\mathcal{X}}^2} \exp(\psi^\top(g_\omega([x^1, \tilde{x}^2]))) \right],$$

where we sample negative sentences from the corpus and combine it with the positive sentence to construct $\tilde{\mathcal{X}}^2$. To make the connection of this objective function with InfoNCE in Eq. 1 explicit, let $a$ and $b$ be two consecutive sentences $x_1$ and $x_2$. Let $f_\theta(a, b)$ be $\psi^\top g_\omega([a, b])$, where $\psi \in \mathbb{R}^d$ is a trainable parameter, $[a, b]$ denotes a concatenation of $a$ and $b$. Consider a Transformer encoder parameterized by $\omega$, and let $g_\omega([a, b]) \in \mathbb{R}^d$ be a function that returns the final hidden state of the first token after running the concatenated sequence to the Transformer. Note that the encoder that we want to learn only depends on $g_\omega$, so both of these approximations can be used for training next sentence prediction.

## B    HYPERPARAMETERS

**Pretraining.**    We use Adam (Kingma & Ba, 2015) with $\beta_1 = 0.9$, $\beta_2 = 0.98$ and $\epsilon = 1e - 6$. The batch size for training is 1024 with a maximum sequence length of 512. We train for 400,000 steps (including 18,000 warmup steps) with a weight decay rate of 0.01. We set the learning rate to $4e^{-4}$ for all variants of the BASE models and $1e^{-4}$ for the LARGE models. We set $\lambda_{\text{MLM}}$ to 1.0 and tune $\lambda_{\text{DIM}} \in \{0.4, 0.6, 0.8, 1.0\}$.

**GLUE.**    We set the maximum sequence length to 128. For each GLUE task, we use the respective development set to choose the learning rate from $\{5e^{-6}, 1e^{-5}, 2e^{-5}, 3e^{-5}, 5e^{-5}\}$, and the batch size from $\{16, 32\}$. The number of training epochs is set to 4 for CoLA and 10 for other tasks, following Joshi et al. (2019). We run each hyperparameter configuration 5 times and evaluate the best model on the test set (once).

**SQuAD.** We set the maximum sequence length to 512 and train for 4 epochs. We use the development set to choose the learning rate from $\{5e^{-6}, 1e^{-5}, 2e^{-5}, 3e^{-5}, 5e^{-5}\}$ and the batch size from $\{16, 32\}$.

## C   QUESTION ANSWERING DECODER

We use a standard span predictor as follows. Denote the length of the context paragraph by $M$, and $\boldsymbol{x}^{\text{context}} = \{x_1^{\text{context}}, \ldots, x_M^{\text{context}}\}$. Denote the encoded representation of the $m$-th token in the context by $\mathbf{x}_{t,m}^{\text{context}}$. The question answering decoder introduces two sets of parameters: $\mathbf{w}_{\text{start}}$ and $\mathbf{w}_{\text{end}}$. The probability of each context token being the start of the answer is computed as: $p(\texttt{start} = x_{t,m}^{\text{context}} \mid \boldsymbol{x}_t) = \frac{\exp(\mathbf{w}_{\text{start}}^{\top}\mathbf{x}_{t,m}^{\text{context}})}{\sum_{n=0}^{M} \exp(\mathbf{w}_{\text{start}}^{\top}\mathbf{x}_{t,n}^{\text{context}})}$. The probability of the end index of the answer is computed analogously using $\mathbf{w}_{\text{end}}$. The predicted answer is the span with the highest probability after multiplying the start and end probabilities.

