# OpenReview forum: "A Mutual Information Maximization Perspective of Language Representation Learning"
_ICLR.cc/2020/Conference — Accept (Spotlight)_

### Official Review · AnonReviewer1 · 2019-10-16
**Official Blind Review #1**

**Rating:** 8

**Review:**

The paper gives a big picture view on training objectives used to obtain static and contextualized word embeddings. This is very handy since classical static word embeddings, such as SGNS and GloVe, have been studied theoretically in a number of works (e.g., Levy and Goldberg, 2014; Arora et al., 2016; Hashimoto et al., 2016; Gittens et al., 2017; Allen and Hospedales, 2019; Assylbekov and Takhanov, 2019), but not much has been done for the modern contextualized embedding models such ELMo and BERT - I personally know only the work of Wang and Cho (2019), and please correct me if I am wrong.

"There is nothing as practical as a good theory", and the authors confirm this statement: their theory suggests them to modify the training objective of the masked language modeling in a certain way and this modification proves to benefit the embeddings in general when evaluated on standard tasks.

I don't have any major issues to raise. A minor comment is that the mutual information I(., .) being a function of two variables suddenly became a function of a single variable in Eq. (1) and in the text which precedes it.

**Experience Assessment:**

I have published in this field for several years.

**Review Assessment: Checking Correctness Of Derivations And Theory:**

I carefully checked the derivations and theory.

**Review Assessment: Checking Correctness Of Experiments:**

I assessed the sensibility of the experiments.

**Review Assessment: Thoroughness In Paper Reading:**

I read the paper at least twice and used my best judgement in assessing the paper.

---

> ### Author Response · Authors · 2019-11-08
> **response**
>
> Thank you for your thoughtful review. We have updated Equation 1 and the paragraph above so that I(...) is consistently a function of two variables.

---

### Official Review · AnonReviewer3 · 2019-10-23
**Official Blind Review #3**

**Rating:** 8

**Review:**

The paper proposes to make a clear connection between the InfoNCE learning objective (which is a lower bound of the mutual information) and multiple language models like BERT and XLN. Then based on the observation that classical LM can be seen as instances of InfoNCE, they propose a new (InfoWord) model relying on the same principles, but taking inspiration from other models also based on InfoNCE. Mainly, the proposed model  differs both in the nature of the a and b variables used in InfoNCE, and also on the fact that it uses negative sampling instead of softmax. Experiments are made on two tasks and compared to a classical BERT model, and on the BERT-NCE model that is a BERT variant proposed by the authors which is somehow in-between BERT and InfoWord. They show that their approach works quite well.

I have a very mitigated opinion on the paper. I) First, I really like the idea of trying to unify different models under the same learning principles, and then show that these models can be seen as specific instances of generic principles. But the way it is presented and explained lacks of clarity: for instance in Section 2, some notations are not well defined (e.g what is f?) . Moreover, the way classical models are casted under the InfoNCE principle is badly written: it assumes that readers have a very good knowledge of the models, and the paper does not show well the mapping between the loss function of each model and the InfoNCE criterion. It gives technical details that could (in my opinion) get ignored, and I would clearly prefer to catch the main differences between the different models that being flooded by technical details. So, my suggestion would be to improve the writing of this section to make the message stronger and relevant for a larger audience. II) The Infoword model can be seen as a simple instance of word masking based models, and as an extension of deep infomax for sequences (it would be certainly nice to describe a little bit what Deep InfoMax is to facilitate the reading).  Here again, the article moves from technical details (e.g "hidden state of the first token (assumed to be a special start of sentence symbol ") without providing formal definitions. Having a first loss function after paragraph 4 could help to understand the principle of this model (before restricting the model to n-grams).  Moreover, the equation J_DIM seems to be wrong since it contains g_\omega twice while I think (but maybe I am wrong) that it has also to be defined by g_\psi. J_MLM is also not clear since x_i is never defined (I assume it is x_{i:i}). At last,  after unifying multiple models under one common learning objective, the authors propose to mix two different losses which is strange (the effect of the second term is slightly studied in the experimental section) without allowing us to understand why it is important to have this second loss function and why the first one is not sufficient enough. At last, I am pretty sure to not be able to reproduce the model described in the paper (adding a section on that in the supplementary material would help), and many concrete aspects are described too fast (like the way to sample negative pairs).

Concerning the experimental section, experiments are convincing and show that the model is able to achieve a performance which is close to classical models. In my opinion, tis section has to be interpreted as  a proof that the proposed unified vision is a good way to easily define new and efficient models.

To summarize, the unification under the InfoNCE principle is interesting,  but the way the paper is written makes it very difficult to follow, and the description of the proposed model is unclear (making the experiments difficult to reproduce) and lacks of a better discussion about the interest of mixing multiple loss.




**Experience Assessment:**

I have published in this field for several years.

**Review Assessment: Checking Correctness Of Derivations And Theory:**

N/A

**Review Assessment: Checking Correctness Of Experiments:**

I assessed the sensibility of the experiments.

**Review Assessment: Thoroughness In Paper Reading:**

I read the paper thoroughly.

---

> ### Author Response · Authors · 2019-11-08
> **response**
>
> Thank you for your thoughtful review.
>
> We have updated the paper based on your comments to improve clarity and reproducibility. We list a summary of our main changes below:
> - In order to make it easier for readers to understand the differences between different models and how they are related to InfoNCE, we have added a summary in Table 1.
> - We have improved notations by adding explicit definitions before they are used in Section 2 and Section 4, and added a short description of Deep InfoMax in Section 4.
> - We have included model and training hyperparameter details in Section 5.1 and Appendix B.
> - We added a motivation for mixing two different terms in the objective function. Our DIM is primarily designed to improve sentence and span representations. We combine it with MLM which is designed for learning (contextual) word representations, since our overall goal is to create better representations for both the sentence and each word in the sentence. We also note that Deep InfoMax for learning image representations mixes multiple terms in their objective function. We only take one of the terms from the full objective function and mix it with MLM.
>
> Regarding equation I_{DIM}, it is supposed to contain two g_{\omega} and no g_{\psi} as we use one network for encoding both the sentence and n-grams. This is not a typo.

---

### Official Review · AnonReviewer2 · 2019-10-26
**Official Blind Review #2**

**Rating:** 6

**Review:**

This paper first gives a concise yet precise summary of maximizing one of variational lower bounds of mutual information, InfoNCE, then it provides an alternative view to explain case by case why word embedding Skip-gram, BERT, XLNet work in practice can be viewed by InfoNCE framework, thus we have a good understand for these methods. Moreover it introduces a self-learning method  that maximizes the mutual information between a global sentence representation and n-grams in the sentence based on deep InfoMax framework instead. Experiments show that it is better then BERT and BERT-NCE. It's known that InfoNCE increases bias but reduce variance, the same is true for deep InfoMax. Do you observe this in your experiments? If so, please provide.

The paper is well-written and easy to follow. The originality is relative low though, since it is mainly an application of  deep InfoMax to language modeling, not inventing a new algorithm and applying to language modeling.

In equations 1 and 2, should a, b be written in capital? Since they represent random variables.

**Experience Assessment:**

I have read many papers in this area.

**Review Assessment: Checking Correctness Of Derivations And Theory:**

I carefully checked the derivations and theory.

**Review Assessment: Checking Correctness Of Experiments:**

I carefully checked the experiments.

**Review Assessment: Thoroughness In Paper Reading:**

I read the paper thoroughly.

---

> ### Author Response · Authors · 2019-11-08
> **response**
>
> Thank you for your thoughtful review.
>
> We have updated notations in Equations 1 and 2. The expectations are now taken over random variables (A and B) and the function takes particular values (a and b) of these random variables.
>
> Regarding your comment about increasing bias and reducing variance, we did observe that the quality of the InfoWord representations is relatively stable across different runs in our experiments (as evaluated by performance on downstream tasks). Could you please clarify a bit more whether this is what you are asking?

---

> > ### Comment · AnonReviewer2 · 2019-11-15
> > **Official Blind Review #1**
> >
> > Yes, it is.

---

> > > ### Author Response · Authors · 2019-11-15
> > > **response**
> > >
> > > Thank you for the clarification. I hope we have answered your question above.
> > >
> > > Regarding novelty, the main contribution of the paper is a unifying framework of language representation learning models based on mutual information maximization. The framework also allows us to easily construct new self-supervised tasks and take inspirations from similar methods that have been successful in other domains. We use Deep InfoMax as an example to validate this claim, but training objectives derived from other methods such as AMDIM and CPC are also possible.
> > >
> > > Please let us know if you have any other questions or concerns, and thank you for helping us improve the submission.

---

### Public Comment · ~Zhengyan_Zhang1 · 2019-12-23
**Questions about I_{DIM} and I_{MLM}.**

Dear authors,

Thank you for the interesting paper and congratulations on the acceptance at ICLR.

After reading, I have two questions about I_{DIM} and I_{MLM}:
1. According to the paper, \hat{x_{i:j}} in I_{DIM} is a sentence masked at position i to j. I wonder whether \hat{x_{i:j}} follows the masking budget (15% of the sequence length) and there are several masked n-grams in this sentence.
2. In I_{MLM}, does g_{\psi} give the contextualized word embeddings from the final layer? If so, I think the model should compute the masked sentence for g_{\omega} and unmasked sentence for g_{\psi} simultaneously.

---

> ### Author Response · Authors · 2020-02-14
> **reply**
>
> Thanks for the comments.
>
> For 1, yes. The \hat{x_{i:j}} follows the masking budget (15% of the sequence length) and there are several masked n-grams in this sentence.
>
> For 2, in MLM g_{\psi} is a simple lookup same as in the original BERT.

---

### Public Comment · ~Martin_Ma1 · 2020-02-24
**Code for this paper**

Dear authors,

Greetings!

May I ask will the code for this paper be possibly public?

Thank you so much!

---

### Decision · Program_Chairs · 2019-12-19

**Decision:**

Accept (Spotlight)

**Comment:**

This paper explores several embedding models (Skip-gram, BERT, XLNet) and describes a framework for comparing, and in the end, unifying them.  The framework is such that it actually suggests new ways of creating embeddings, and draws connections to methodology from computer vision.

One of the reviewers had several questions about the derivations in your paper and was worried about the paper's clarity.  But all of the reviewers appreciated the contributions of the paper, which joins multiple seemingly disparite models under into one theoretical framework.

The reviewers were positive about the paper, and in particular were happy to see the active response of authors to their questions and willingness to update the paper with their suggested improvements.